# CodeXEmbed: A Generalist Embedding Model Family for Multilingual and Multi-task Code Retrieval

Ye Liu, Rui Meng,* Shafiq Joty, Silvio Savarese, Caiming Xiong, Yingbo Zhou, Semih Yavuz
Salesforce AI Research
{yeliu, yingbo.zhou, syavuz}@salesforce.com

## Abstract

Despite the success of text retrieval in many NLP tasks, code retrieval has received comparatively less attention than text retrieval. Most retrieval systems are designed for natural language queries and often fail to address the structural and semantic complexities of retrieving code. Consequently, they struggle across diverse programming languages and retrieval tasks, underscoring the need for specialized models tailored to code retrieval. To address this gap, we introduce CODEXEM-BED, a family of large-scale code embedding models ranging from 400M to 7B parameters. Our novel training pipeline integrates multiple programming languages and reformulates diverse code-related tasks within a unified retrieval framework, enhancing model generalizability and performance. Our largest model (7B parameters) sets a new state-of-the-art (SOTA) in code retrieval, ranking first on the CoIR Leaderboard. Beyond code retrieval, our models achieve competitive results on the widely adopted BeIR text retrieval benchmark, demonstrating cross-domain versatility. Furthermore, our findings highlight that advancements in retrieval quality directly improve end-to-end Retrieval-Augmented Generation (RAG) performance for code-related tasks.

## 1 Introduction

Large Language Models (LLMs) have demonstrated exceptional performance across numerous Natural Language Processing (NLP) tasks. However, they often struggle to produce faithful answers and may lack up-to-date or domain-specific knowledge. To bridge this gap, retrieval-augmented generation (RAG) Cai et al. (2022); Cheng et al. (2024); Nguyen et al. (2024) techniques have gained prominence, integrating Information Retrieval (IR) systems with LLMs to enhance their access to relevant external information. This synergy has drawn significant attention recently, leading to the development of various retrieval models Wang et al. (2022); Chen et al. (2024) based on BERT Kenton & Toutanova (2019) and other LLMs with sizes exceeding 1 billion parameters Wang et al. (2023); Moreira et al. (2024); Meng et al. (2024). Despite these advancements, standard IR methods, while effective in text-based retrieval, often fall short in specialized domains such as code retrieval Husain et al. (2019); Li et al. (2024b).

Code retrieval accelerates software development and improves code quality by enabling quick access to relevant snippets, explanations, and analyses. Unlike general text retrieval, it must handle syntax, dependencies, and control flow. Integrated into tools like VS Code Del Sole (2021) and GitHub Copilot Wermelinger (2023); Yetiştiren et al. (2023), code retrieval also enhances Code-RAG systems Parvez et al. (2021); Liu et al. (2020); Jimenez et al. (2024); Wang et al. (2024) by reducing LLM hallucinations. However, traditional text retrieval models struggle with code-specific elements. Existing models like CodeBERT Feng et al. (2020), CodeGPT Lu et al. (2021), and UniXcoder Guo et al. (2022) are based on smaller BERT models Kenton & Toutanova (2019) leading to subpar performance. Furthermore, a unified model capable of handling both text and code retrieval is essential for seamless integration, enabling developers to retrieve documentation, explanations, and relevant code within a single framework.

---

*Now at Google.

In this work, we introduce CODEXEMBED[1], a family of open-source embedding models tailored for both code and text, available in sizes of 400 million, 2 billion, and 7 billion parameters. CODEX-EMBED introduces a generalizable approach that converts diverse code-related tasks with various programming languages into a unified contrastive training framework. Our approach handles 12 programming languages and five distinct code retrieval categories across eight different code tasks, including code-to-text, text-to-code, code-to-code, text-to-text and hybrid text and code tasks. To enable the model to retrieve both text and code effectively, we propose a novel multi-stage training method. In the first stage, we train the model on text data using LoRA adaptation. In the second stage, we continue training with both code and text data while tuning only the newly introduced LoRA adapter. This approach preserves the text retrieval knowledge learned in the first stage while enhancing code retrieval, ensuring the model retains strong performance across both domains. This comprehensive setup enables CODEXEMBED to generalize effectively across various code domains. Our contributions can be summarized as follows:

- We introduce a generalizable multi-stage training approach for code and text embedding, unifying diverse code-related tasks within a retrieval framework. This leads to significant improvements in retrieval performance across multiple programming languages and tasks.
- Our 7B model achieves state-of-the-art performance on the CoIR benchmark, setting a new standard for code retrieval. We furhter evaluate CODEXEMBED on RepoEval and SWE-Bench-Lite, demonstrating that improved retrieval significantly enhances end-to-end Retrieval-Augmented Generation (RAG) performance for code-related tasks.
- Beyond the 7B model, we develop smaller models (400M, 2B) that surpass prior SOTA in code retrieval, while remaining competitive performance in text retrieval, highlighting their versatility across both domains.

## 2 Method

We transform general code-related tasks into a unified retrieval framework by representing each task as a structured retrieval pair $(Q, D)$, where $Q$ is the query and $D$ is the positive document. The retrieval process is formulated as a ranking problem, where the objective is to learn a function $f(Q, D)$ that assigns higher similarity scores to relevant pairs than that of irrelevant ones.

### 2.1 Unified Retrieval Framework

To train a unified retrieval model for both code and text, we transform all retrieval tasks into a query-document matching problem, where a query $Q$ retrieves a relevant document $D$ from a candidate set $\mathcal{D}$. This formulation supports Text-to-Text, Text-to-Code, Code-to-Text, and Code-to-Code retrieval within a single framework.

To better capture semantic relationships across entire input sequences, we incorporate *bidirectional attention* into a causal LLM BehnamGhader et al. (2024), enhancing its ability to encode contextual information for retrieval tasks. Each query $Q$ and document $D$ are encoded using this pre-trained LLM, producing representations:

$$q = \text{LLM}_{\theta + \Delta\theta}(Q), \quad d = \text{LLM}_{\theta + \Delta\theta}(D)$$

The similarity between query and document is measured using cosine similarity:

$$D^* = \arg\max_{D \in \mathcal{D}} \text{sim}(q, d), \quad \text{sim}(q, d) = \frac{q \cdot d}{\|q\| \|d\|} \tag{1}$$

### 2.2 Multi-Stage Training with LoRA

To progressively enhance retrieval capabilities, we adopt a multi-stage training approach, utilizing distinct LoRA Hu et al. (2022) adapters at each stage. In each stage, the previously trained LoRA weights are merged into the base model before initializing a new adapter, as shown in Figure 1. This approach enables the model to iteratively adapt to different retrieval objectives without catastrophic forgetting Magistri et al. (2024); Han et al. (2024)

---

[1]Model weights: https://huggingface.co/Salesforce/SFR-Embedding-Code-2B_R, https://huggingface.co/Salesforce/SFR-Embedding-Code-400M_R.

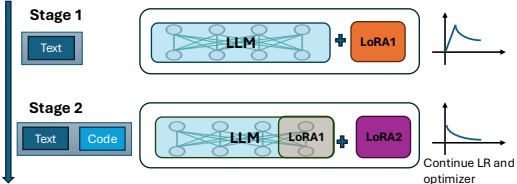

Figure 1: The illustration depicts multi-stage training, showing the data used at each stage, the LoRA adapters applied, and the continuous optimizer with its learning rate progression.

### 2.2.1 Stage 1: Supervised Contrastive Pretraining on Text Data

We initialize a LoRA adapter $\Delta\theta_1$ and freeze the LLM $\theta$ and train $\Delta\theta_1$ on text-based retrieval tasks using a contrastive learning objective:

$$\mathcal{L} = -\frac{1}{N} \sum_{i=1}^{N} \log \frac{\exp(\text{sim}(q_i, d_i^+))}{\exp(\text{sim}(q_i, d_i^+)) + \sum_{j=1}^{K} \exp(\text{sim}(q_i, d_j^-))} \tag{2}$$

where $d_i^+$ is a positive document, and $d_j^-$ represents negative documents, and $N$ denotes the batch size. The model learns to align text queries with their corresponding text-based documents, establishing a strong foundation for language-based retrieval.

### 2.2.2 Stage 2: Code-Enhanced Retrieval Training

After training, we merge the trained adapter $\Delta\theta_1$ into the base LLM model:

$$\theta \leftarrow \theta + \Delta\theta_1 \tag{3}$$

and initialize a new LoRA adapter $\Delta\theta_2$ to learn retrieval for both code and text. This setup enables the model to mitigate text-only biases while preserving most of its learned representations.

To ensure a smooth transition between training phases, we continue optimizing the second-stage LoRA adapter $\Delta\theta_2$ while retaining the gradient contributions from the first-stage adapter $\Delta\theta_1$, which remains fixed within the model. Specifically, we update $\Delta\theta_2$ while preserving the learning rate $r^{(1)}$ and the accumulated gradient from $\Delta\theta_1$:

$$\Delta\theta_2 \leftarrow \text{Update}(\Delta\theta_2, \nabla L(\theta_1, \theta_2), r^{(1)}) \tag{4}$$

where $\nabla L(\theta_1, \theta_2)$ represents the gradient influenced by both adapters, but only $\Delta\theta_2$ is actively updated. This approach stabilizes optimization, preventing abrupt parameter shifts while allowing a smooth adaptation to code retrieval. By preserving the previous learning rate and gradient continuity, we mitigate convergence instability and facilitate efficient adaptation.

This multi-stage approach enables the model to iteratively improve retrieval performance while preventing overfitting to any single modality. By leveraging LoRA adapters at different stages and maintaining a smooth learning rate transition, we ensure efficient adaptation with minimal overhead, resulting in more robust text and code retrieval.

### 2.3 Scaling Batch Size with GradCache

Larger batch sizes improve contrastive learning by incorporating more diverse negatives Chen et al. (2022), but GPU memory limits expansion. To overcome this, we use *GradCache* Gao et al. (2021), which reduces memory overhead, enabling larger effective batch sizes. Let $Q/D$ denote the full batch of queries and documents, partitioned into sub-batches ($Q = \{\hat{Q}_1, \hat{Q}_2, \ldots, \hat{Q}_M\}$, $D = \{\hat{D}_1, \hat{D}_2, \ldots, \hat{D}_M\}$) to fit memory constraints. Training follows three steps:

*Graph-less Forward Pass* Each sub-batch is encoded into embeddings $q_i$ and $d_i$ without computing encoder gradients, reducing memory usage.

*Gradient Computation and Caching* For each query $q_i$ in sub-batch $\hat{Q}_j$, we compute and cache gradients w.r.t. the representation function:

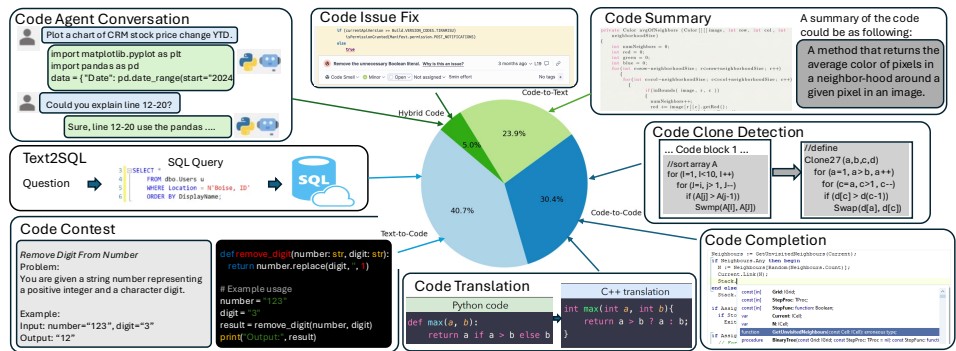

Figure 2: The code training data of CODEXEMBED contains four parts: Text-to-Code, Code-to-Code, Code-to-Text and Hybrid Code. Each Categories contains several types of code tasks.

$$\mathbf{u}_i = \frac{\partial \mathcal{L}}{\partial q_i}, \quad \mathbf{v}_i = \frac{\partial \mathcal{L}}{\partial d_i} \tag{5}$$

*Sub-batch Gradient Accumulation* We reprocess sub-batches to collect embeddings $q_i$ and $d_i$, construct the computation graph, and link gradients:

$$\frac{\partial \mathcal{L}}{\partial \theta} = \sum_{\hat{Q}_j \in Q} \sum_{q_i \in \hat{Q}_j} \mathbf{u}_i \frac{\partial q_i}{\partial \theta} + \sum_{\hat{D}_j \in D} \sum_{d_i \in \hat{D}_j} \mathbf{v}_i \frac{\partial d_i}{\partial \theta} \tag{6}$$

$\partial \theta$ denotes the LoRA adapter in either stage 1 or 2. GradCache enables larger in-batch negatives, improving contrastive learning efficiency while staying within GPU limits, essential for scaling text and code retrieval.

## 2.4 Unified Retrieval Training Data

We organize retrieval tasks into distinct settings by converting various code-related tasks into a unified retrieval format. In this format, the input is treated as *query* ($Q$), and the expected output or relevant content is treated as *document* ($D$). This transformation allows the model to generalize across various retrieval tasks in both text and code domains.

### 2.4.1 Text Retrieval Training Data

To improve retrieval across both code and text domains, we incorporate text-to-text retrieval tasks from the BEIR datasets Thakur et al. (2021). This ensures model robustness in general document retrieval while maintaining strong performance in code retrieval.

### 2.4.2 Code Retrieval Training Data

We unify multiple code generation and classification tasks into the $(Q, D)$ format for code retrieval training, as shown in Figure 2. **Text-to-Code Retrieval** involves retrieving relevant code snippets given a natural language query. Tasks such as *code contest generation* Billah et al. (2024); Kadir et al. (2024) take problem descriptions as $Q$ and map them to the correct implementation as $D$. Similarly, in *Text-to-SQL* Finegan-Dollak et al. (2018); Li et al. (2024a), a user query in natural language serves as $Q$, and the corresponding SQL query is treated as $D$. **Code-to-Text Retrieval** retrieves human-readable descriptions for given code snippets. In *code summarization* Sontakke et al. (2022); Sun et al. (2024), a code snippet serves as $Q$, and the corresponding documentation or summary is $D$. **Code-to-Code Retrieval** represents retrieving functionally equivalent or semantically related code snippets. In *code translation* Pan et al. (2024); Karanjai et al. (2024), the source code in one language is $Q$, and its equivalent implementation in another language is $D$. *Code completion* Ding et al. (2024); Phan et al. (2024); Liu et al. (2024) uses the incomplete code fragment as $Q$ and the

correct completion as $D$. *Code clone detection* Martinez-Gil (2024) retrieves functionally similar code, treating the reference snippet as $Q$ and the detected duplicate as $D$. **Hybrid Code Retrieval** supports mixed queries containing both text and code. In *code agent conversation* Arteaga Garcia et al. (2024); Jin et al. (2024), user prompts containing explanations or code snippets act as $Q$, while the retrieved response (either code or text) serves as $D$. In *code issue fixing* Yang et al. (2024); Jimenez et al. (2024), the hybrid query consists of an error message and buggy code as $Q$, while the corrected version of the code is $D$.

By consolidating diverse retrieval tasks into a unified framework, we effectively streamline contrastive model training, thereby facilitating seamless adaptation across a wide range of programming scenarios while ensuring high retrieval accuracy.

## 3 Experiments

**Evaluation Benchmarks** We mainly use two benchmarks to evaluate code and text retrieval performance. **COIR** Li et al. (2024b) is a comprehensive benchmark specifically designed for code retrieval tasks. COIR covers a wide range of retrieval challenges, including 8 fine-grained retrieval subtasks, spanning 14 major programming languages. The dataset is composed of 10 distinct datasets, with a total corpus exceeding 2 million entries. **BEIR** Thakur et al. (2021) is a widely-adopted benchmark designed for text retrieval tasks. BEIR encompasses a diverse set of retrieval challenges, covering 9 distinct tasks across various domains such as question answering, duplicate detection, fact-checking, and more. It supports retrieval over a wide range of datasets and provides a standardized benchmark for evaluating text retrieval models across different domains.

**Implementation Details** We conduct general training on our proposed code and text pair dataset using three model sizes: 400M, 2B, and 7B. For the CodeXEmbed$_{400M}$, we use the base model *Alibaba-NLP/gte-large-en-v1.5* (Li et al., 2023b), applying full model fine-tuning. For the CodeXEmbed$_{2B}$, we initialize our embedding model from the generation model *google/gemma-2-2b-it* Team et al. (2024), using low-rank adaption(LoRA) Hu et al. (2022) with a rank of 8. For the CodeXEmbed$_{7B}$, we initialize our embedding model from the generation model *mistralai/Mistral-7B-Instruct-v0.3*, also using LoRA with a rank of 8. Following prior work Meng et al. (2024), we apply: (i) last token pooling for CodeXEmbed$_{2B}$ and CodeXEmbed$_{7B}$, and (ii) beginning token pooling for CodeXEmbed$_{400M}$ to generate semantic vector representations Li et al. (2023b). Cosine similarity is used to compute the similarity between query and corpus for ranking. The batch size is set to 1024 across all three model sizes, with 7 hard negatives. The learning rate is set to $5e^{-5}$, and the end learning rate to $5e^{-6}$, with linear decay and a 50-step warmup. To improve training efficiency and reduce GPU memory usage, we adopt gradient caching Gao et al. (2021). The more implementation details can be found in Appendix A.3.

**Baseline Models** For code-domain-specific models, we included UniXcoder (Guo et al., 2022), Voyage-Code-002[2] and CodeSage-large-v2 (Zhang et al., 2024), all pre-trained on code data, serving as strong baselines for comparison. For general retrieval models, we evaluated E5-Base (Wang et al., 2022), GTE-Base (Li et al., 2023b), BGE-Base (Xiao et al., 2023), Contriever (Izacard et al., 2023), E5-Mistral (Wang et al., 2023), BGE-M3 (Chen et al., 2024), NV-Embed-V2 Moreira et al. (2024), SFR-V2 Meng* et al. (2024) and OpenAI-Ada-002[3].

**Evaluation Metrics** In code retrieval, selecting the right metric is key for assessing both ranking sensitivity and relevance. Following prior work Wang et al. (2013), Normalized Discounted Cumulative Gain (NDCG) is preferred for its ability to account for both rank order and varying relevance. Therefore, we use NDCG@10 to evaluate performance on CoIR[4] and BEIR.

### 3.1 General Training Evaluation

In the *General Training* block of Table 1, we present the results of models trained exclusively on our proposed general training data, without using any CoIR in-domain data. When averaged over all 10 datasets in the CoIR benchmark, CodeXEmbed$_{7B}$ model achieves the best results, exceeding the

---

[2] https://blog.voyageai.com/2024/01/23/voyage-code-2-elevate-your-code-retrieval/
[3] https://platform.openai.com/docs/guides/embeddings
[4] CoIR Implementation https://github.com/CoIR-team/coir

| Model | Text-to-Code | | | Code-to-Text | Code-to-Code | | | Hybrid Code | | | Avg |
|---|---|---|---|---|---|---|---|---|---|---|---|
| | Apps | CosQA | Text2SQL | CSN | CSN-CCR | CodeTrans-Contest | -DL | StackOverFlow QA | CodeFeedBack-ST | -MT | |
| *Baselines* | | | | | | | | | | | |
| E5-base (110M) | 11.52 | 32.59 | 52.31 | 67.99 | 56.87 | 62.50 | 21.87 | 86.86 | **74.52** | 41.99 | 50.90 |
| BGE-Base (110M) | 4.05 | **32.76** | 45.59 | 69.60 | 45.56 | 38.50 | 21.71 | 73.55 | 64.99 | 31.42 | 42.77 |
| UniXcoder (123M) | 1.36 | 25.14 | 50.45 | 60.20 | 58.36 | 41.82 | 31.03 | 44.67 | 36.02 | 24.21 | 37.33 |
| BGE-M3 (567B) | 7.37 | 22.73 | 48.76 | 43.23 | 47.55 | 47.86 | 31.16 | 51.04 | 49.94 | 33.46 | 39.31 |
| E5-Mistral (7B) | 21.33 | 31.27 | 65.98 | 54.25 | 65.27 | 82.55 | 33.24 | **91.54** | 72.71 | 33.65 | 55.18 |
| OpenAI-Ada-002 | 8.70 | 28.88 | 58.32 | 74.21 | 69.13 | 53.34 | 26.04 | 72.40 | 47.12 | 17.74 | 45.59 |
| Voyage-Code-002 | 26.52 | 29.79 | **69.26** | 81.79 | 73.45 | 72.77 | 27.28 | 67.68 | 65.35 | 28.74 | 56.26 |
| CodeSage-large-v2 | 50.45 | 32.73 | 59.78 | **94.26** | 78.09 | 85.27 | 33.29 | 79.41 | 71.32 | **57.16** | 64.18 |
| *General Training* | | | | | | | | | | | |
| CodeXEmbed400M | 48.57 | 34.05 | 58.96 | 72.53 | 80.15 | 75.67 | **34.85** | 89.51 | 78.87 | 45.75 | 61.89 |
| CodeXEmbed2B | 74.99 | **36.31** | 59.00 | 73.50 | 85.77 | 86.63 | 33.17 | 90.54 | **81.15** | 53.08 | 67.41 |
| CodeXEmbed7B | 85.22 | 33.27 | 64.57 | **78.84** | 86.77 | 90.64 | 32.31 | 94.25 | 80.93 | 57.83 | 70.46 |
| *In-domain Training* | | | | | | | | | | | |
| CodeXEmbed400M | 45.91 | 41.28 | 61.29 | 81.23 | 93.74 | 82.72 | **40.81** | 92.35 | 83.36 | 61.51 | 68.42 |
| CodeXEmbed2B | 76.86 | 40.47 | 78.42 | 87.87 | 97.66 | 90.30 | 38.57 | 94.47 | 86.36 | 65.51 | 75.65 |
| CodeXEmbed7B | **85.38** | **42.47** | **78.94** | 89.67 | **97.95** | **94.45** | 40.46 | **96.33** | **87.53** | **68.83** | **78.20** |

Table 1: Performance of the CODEXEMBED model family across tasks, along with average scores. CSN refers to CodeSearchNet. Baseline numbers are from the CoIR Leaderboard.

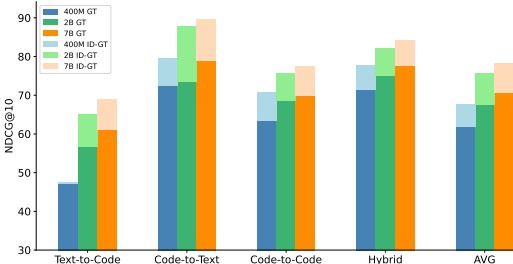

Figure 3: The performance comparison between General Training (GT) and In-domain Training (ID) across three model sizes (400M, 2B, and 7B) on different CoIR categories and the overall average.

SOTA code-domain specific model Voyage-Code-002 by over 20%[5], which shows our general code and text training stage significantly improves model performance on code tasks.

As shown in Table 1, CodeXEmbed400M and CodeXEmbed2B also provide a significant improvement over Voyage-Code-002 and offer a great alternative to the 7B model with substantial practical advantages on the latency and cost. Moreover, their success further validates the transferability and generalizability of our proposed training recipe for code embedding models.

### 3.2 In-domain Training Evaluation

We further trained the model on the CoIR in-domain dataset. As shown in the *In-domain Training* block of Table 1, further training on in-domain data results in consistent performance improvements across all model sizes. Specifically, CodeXEmbed400M improves by 6.5 points, CodeXEmbed2B by 8.24 points, and CodeXEmbed7B by 7.74 points on average across all 10 datasets. In Figure 3, the top of each bar represents the improvement from in-domain training. Among all categories, the Code-to-Text category shows the largest improvement, even outperforming Voyage-Code-002. In other categories, the model also achieves over a 5-point improvement.

### 3.3 Unified Text and Code Retrieval

To evaluate the text and code retrieval capabilities within a single embedding model, we also present the BEIR text retrieval performance of CODEXEMBED across various sizes. As shown in Table 2, our 7B model achieves an average score of over 60 across 15 datasets, placing it among the top tier on the MTEB leaderboard[6]. Compared to E5-Mistral-7B-Instruct (E5-Mistral) Jiang et al. (2023), which

---

[5]Baseline numbers are sourced from the CoIR Leaderboard: https://archersama.github.io/coir/.

[6]https://huggingface.co/spaces/mteb/leaderboard

| Dataset | BM25 | gte-large 400M | gte-Qwen2 1.5B | E5-Mistral 7B | CODEXEMBED 400M | CODEXEMBED 2B | CODEXEMBED 7B |
|---|---|---|---|---|---|---|---|
| MS MARCO | 22.8 | 42.93 | **43.36** | 43.06 | 42.77 | 41.26 | 42.05 |
| TREC-Covid | 65.6 | 77.49 | 85.38 | **87.03** | 77.47 | 84.58 | 79.04 |
| NFCorpus | 32.5 | 36.95 | 39.34 | 38.58 | 35.76 | 41.56 | **43.14** |
| NQ | 32.9 | 56.08 | 56.08 | 63.53 | 63.38 | 67.25 | **74.11** |
| HotpotQA | 60.3 | 68.18 | 64.00 | 75.72 | 74.93 | 74.39 | **79.33** |
| FiQA | 23.6 | **63.23** | 63.23 | 56.81 | 60.20 | 56.17 | 60.41 |
| ArguAna | 31.5 | **72.11** | 54.70 | 61.65 | 69.67 | 61.39 | 63.58 |
| Touche-2020 | **36.7** | 22.55 | 27.89 | 26.27 | 20.18 | 26.10 | 25.80 |
| CQADupStack | 29.9 | 42.16 | 44.76 | 42.97 | 46.07 | 47.46 | **51.45** |
| Quora | 78.9 | **89.67** | 89.64 | 89.61 | 89.05 | 89.27 | 89.51 |
| DBPedia | 31.3 | 46.30 | 48.69 | 48.89 | 46.68 | 47.33 | **49.27** |
| Scidocs | 15.8 | **26.35** | 24.98 | 16.32 | 25.05 | 23.36 | 25.25 |
| Fever | 75.3 | 93.81 | 91.57 | 87.84 | **93.86** | 89.03 | 91.94 |
| Climate-Fever | 21.3 | **48.36** | 42.91 | 38.35 | 42.70 | 32.08 | 36.93 |
| Scifact | 66.5 | 82.43 | 78.44 | 76.42 | 87.37 | 84.79 | **89.10** |
| Average | 41.7 | 57.91 | 58.29 | 56.87 | 58.34 | 57.73 | **60.06** |
| Best on | 1 | 5 | 1 | 1 | 1 | 0 | 6 |

Table 2: Comparison of performance across text retrieval BEIR datasets with different model size.

is trained on both text and synthetic data, initialized from the Mistral series, our model employs single-stage training with both code and text data. Our model in a performance improvement of 3.19 points over E5-Mistral.

In the 400M models, CODEXEMBED achieves a 0.43 performance boost over GTE-large (Li et al., 2023b), the model it is trained on. This highlights the advantage of our approach, showing the potential to improve text retrieval by incorporating code data. Few 2B-sized language models are available; we selected Gemma 2B Team et al. (2024) for its strong code retrieval performance. For text retrieval, it performs comparably to gte-Qwen2 of similar size.

| Dataset | NV-Embed-V2 7B | SFR-v2 7B | CODEXEMBED 7B |
|---|---|---|---|
| CoIR | 59.10 | 61.48 | **70.46** |
| BEIR | **62.65** | 60.18 | 60.06 |
| AVG | 60.88 | 60.83 | **65.26** |

Table 3: Comparison of code and text retrieval benchmarks between our model and the top models on the MTEB leaderboard.

We present the performance of top text retrieval models from the MTEB leaderboard[7] on both code and text tasks, in Table 3. Compared to the top-ranked model, NV-Embed-V2 Moreira et al. (2024)[8], our model surpasses it by 4.38 points, with an average score across text and code datasets.

## 3.4 Impact of Multi-Stage Training

To evaluate the effectiveness of our multi-stage training approach, we conduct an ablation study on four settings:

- **Single-stage (Baseline)**: Trained in a single phase using a mixture of text and code retrieval data with a single LoRA adapter.
- **Two-stage (Same LoRA)**: The first stage is trained on text retrieval; the second stage continues with mixed text and code retrieval, reusing the same LoRA adapter across both stages.
- **Two-stage (Different Optimizers)**: Uses different LoRA adapters for each stage, with separate optimizers for each.
- **Proposed Two-stage (Separate LoRA)**: Applies different LoRA adapters for each stage, using a continuous optimizer throughout training.

---

[7]https://huggingface.co/spaces/mteb/leaderboard
[8]Top-performing open model as of 3/28/2025.

| Training Setup | BEIR | CoIR |
|---|---|---|
| Single-stage (Baseline) | 65.79 | 64.56 |
| Two-stage (Same LoRA) | 67.54 | 58.61 |
| Two-stage (Diff. Optimizer) | 66.95 | 66.06 |
| Proposed Two-stage (Separate LoRA) | **67.88** | **67.41** |

Table 4: Comparison of training methods on BEIR dev and CoIR test sets (NDCG@10).

Table 4 shows our Proposed Two-stage (Separate LoRA) achieves the best performance on BEIR (67.88) and CoIR (67.41), highlighting the importance of modular adaptation. The Single-stage (Baseline) struggles on CoIR, while Two-stage (Same LoRA) improves BEIR but drops on CoIR. Two-stage (Different Optimizers) balances performance but falls short of our approach.

## 3.5 Retrieval-Augmented Code Generation

In this section, we explore how different retrievers influence the final code completion and issue resolution performance in repository-level tasks.

### 3.5.1 RepoEval

To address this, we utilize RepoEval (Zhang et al., 2023) for repository-level code completion. While RepoEval consists of three splits (function, class, and module), we report results only on the function split, as it is the only one that supports execution-based evaluation. We adopt Pass@1 as our evaluation metric, which measures the accuracy of the top-1 generated code passing the provided test cases.

For code generation, we supply the top-5 retrieved code snippets as input to the GPT-3.5-turbo generation model. All generator parameters and decoding hyperparameters are kept frozen throughout the experiments. As shown in Table 5, all sizes of CODEXEMBED outperform the canonical setup. While some files may not contain direct solutions, as in the canonical documents, they often include valuable function definitions or usage examples that improve code generation outcomes. This suggests that our embeddings effectively capture the repository structure and retrieve contexts that implicitly support problem-solving.

| Dataset | None | BM25 | Voyage | OpenAI | OpenAI rerank | CODEXEMBED 400M | CODEXEMBED 2B | CODEXEMBED 7B | Gold |
|---|---|---|---|---|---|---|---|---|---|
| `gpt-3.5` | | | | | | | | | |
| **RepoEval** | 23.9 | 30.8 | 43.2 | 48.0 | 49.6 | 52.5 | **66.3** | 63.8 | 39.1 |
| **SWE-Bench-Lite** | 0.7 | 1.0 | 0.7 | 0.3 | 0.0 | 0.7 | 2.0 | **3.0** | 2.7 |
| `gpt-4o` | | | | | | | | | |
| **SWE-Bench-Lite** | 2.3 | - | - | - | 21.7 | 19.7 | 21.7 | **25.0** | 30.7 |

Table 5: Performance of repository-level code retrieval augmented generation using `gpt-3.5-turbo-0125` and `gpt-4o-2024-08-06`. CODEXEMBED variants denote models of size 400M, 2B, and 7B.

### 3.5.2 SWE-Bench-Lite

In our experiments, we use SWE-bench-Lite[9], a curated subset of 300 problems from the original SWE-bench benchmark. It focuses on resolving GitHub issues by requiring models to modify multiple files to pass test cases, offering a manageable and reproducible dataset. SWE-bench-Lite also includes a pre-configured Docker container, ensuring consistent evaluation across systems and further standardizing the testing environment.

We evaluate our retrieval-augmented generation pipeline using GPT-3.5-turbo as the code generator, with all generator parameters and decoding hyperparameters kept frozen. We also conduct parallel

---

[9]https://huggingface.co/datasets/princeton-nlp/SWE-bench_Lite

| | Model Size | Python | Java | Go | PHP | Javascript | Ruby | SQL | AVG | Δ-All |
|---|---|---|---|---|---|---|---|---|---|---|
| | 400M | 59.70 | 72.09 | 79.28 | 70.96 | 70.18 | 73.01 | 58.87 | 69.16 | - |
| All | 2B | 65.73 | 79.40 | 81.01 | 79.39 | 76.76 | 78.42 | 58.23 | 74.14 | - |
| | 7B | **67.56** | 81.36 | **85.76** | **81.89** | 78.37 | **83.11** | **65.74** | **77.68** | - |
| | 400M | 55.43 | 49.57 | 49.56 | 40.57 | 44.57 | 44.57 | 44.21 | 46.93 | −32.1% |
| Python | 2B | 60.86 | 71.46 | 68.69 | 57.88 | 62.95 | 72.21 | 56.09 | 64.31 | −13.3% |
| | 7B | 67.02 | 78.35 | 81.56 | 70.88 | 75.73 | 79.19 | 62.56 | 73.61 | −5.2% |
| | 400M | 54.79 | 78.37 | 72.94 | 70.85 | 69.85 | 68.78 | 50.31 | 66.56 | −3.8% |
| Java | 2B | 63.03 | 82.17 | 79.35 | 80.30 | 76.00 | 76.76 | 58.03 | 73.66 | −0.6% |
| | 7B | 66.24 | **84.25** | 83.72 | 80.18 | **79.82** | 82.82 | 64.59 | 77.37 | −0.4% |

Table 6: Python/Java indicates that only the Python and Java portions were used to train CODEXEM-BED, while All indicates that all programming languages were used. Δ-All represents the difference between the Python/Java AVG score and the All AVG score for the same model size.

experiments with GPT-4o and observe similar trends, confirming that the performance gains from improved retrieval are consistent across model backends.

As shown in Table 5, our results show that incorporating improved retrieval methods significantly enhances end-to-end performance of code retrieval-augmented generation, bringing it closer to using gold content and boosting problem-solving efficiency and accuracy.

## 3.6 Impact of the Base Models

To understand the base model's impact, we examine: (1) if training from a text retrieval model outperforms a generation model, and (2) if a code-specific generation model offers more advantages than a general language model. We present the ablation study comparing an embedding model to a generation LLM in Appendix A.2.1 and the study on code-specific vs. general LLMs in Appendix A.2.2.

## 3.7 Programming Language Transferability

We aim to explore the diversity of programming languages and their unique features. The details of our code training dataset, including language coverage, are provided in Appendix A.1. Our dataset comprises 12 programming languages, with Python representing the highest percentage of the data. For testing, we selected Python and Java due to their distinct programming paradigms: Python is known for its scripting capabilities and Java for its strong object-oriented design. This selection allows us to evaluate our model's performance across a range of programming styles, reflecting the versatility and adaptability of the embedding model.

To better evaluate retrieval performance across languages, we organize the CoIR test set into language-specific experiments by grouping all subsets associated with the same programming language. No additional resplitting is applied. For each language, we include all CoIR test subsets labeled with that language. The details are shown in Table 7.

| Language | CoIR Test Subsets Used |
|---|---|
| Python | APPS · CoSQA · CodeSearchNet-Python · CodeSearchNet-CCR-Python · CodeTrans-DL |
| Java | CodeSearchNet-Java · CodeSearchNet-CCR-Java |
| Go | CodeSearchNet-Go · CodeSearchNet-CCR-Go |
| PHP | CodeSearchNet-PHP · CodeSearchNet-CCR-PHP |
| JavaScript | CodeSearchNet-JavaScript · CodeSearchNet-CCR-JavaScript |
| Ruby | CodeSearchNet-Ruby · CodeSearchNet-CCR-Ruby |
| SQL | Synthetic-Text2SQL |

Table 7: CoIR test subsets used for each programming language.

As shown in Table 6, training with all 12 programming languages yields the best average performance across 7 target languages, compared to training with a single language. However, training on Java-only consistently achieves the highest performance for Java and delivers comparable results to using all languages across all model sizes. For example, the Java-only 7B model scores 77.37, while the all-languages model scores 77.68. When comparing models trained exclusively on Python or Java, the Java-trained model consistently outperforms. This may be because modern language models are

already heavily trained on Python during the generation phase, so relying solely on Python in the retrieval phase may miss important nuances, resulting in suboptimal performance.

## 4 Related Work

### 4.1 Retrieval Models for Text and Code

Text retrieval models have significantly advanced by exploiting supervision from natural language inference tasks and labeled query-document pairs, such as the MS-MARCO passage ranking dataset Bajaj et al. (2016), to train effective text embeddings Izacard et al. (2021); Wang et al. (2022); Xiao et al. (2024). Recently, researchers have leveraged large language models (LLMs) Jiang et al. (2023) as the base for training retrieval models, resulting in state-of-the-art performance. Notable examples include E5-Mistral Wang et al. (2023), SFR-Embedding Meng et al. (2024); Meng* et al. (2024), and NV-Embedding Lee et al. (2024). While numerous models have been developed for text retrieval tasks, few have focused specifically on code retrieval Zhang et al. (2024); Suresh et al. (2024). Among the few are the Voyage code model AI (2024) and OpenAI's embeddings Neelakantan et al. (2022); And embedding for code issue localization Reddy et al. (2025) however, both are closed-source models, limiting their accessibility and adaptability for the wider research community.

### 4.2 Code Retrieval Augmented Generation

Neural code generation has been an important task Lu et al. (2021), and increasingly strong code language models have been developed Roziere et al. (2023); Nijkamp et al. (2023); Li et al. (2023a); Guo et al. (2024); Team (2024) to solve various tasks Chen et al. (2021); Lai et al. (2023); Jimenez et al. (2024). However, most LMs generate code solely from natural language problem descriptions and the models' parametric knowledge, without leveraging external programming resources or using a retrieval-augmented generation approach. While prior work has focused on text-centric tasks with general-domain corpora like Wikipedia Asai et al. (2024), some studies have used retrieved programming context from repositories Ding et al. (2024); Yang et al. (2024) or documentation Zhou et al. (2023). Code retrieval is crucial for enhancing code generation in RAG systems because it allows models to access relevant external code resources, leading to more accurate and context-aware code outputs. Our code retrieval model demonstrates significant improvements in code RAG performance, underscoring the importance of effective code retrieval in code generation tasks.

## 5 Conclusion

In the underexplored field of code retrieval, we present CODEXEMBED, a family of code embedding models ranging from 400M to 7B parameters. Our unified training pipeline integrates multiple programming languages and reformulates diverse code-related tasks within a common retrieval framework. CODEXEMBED ranks as the top model on the CoIR benchmark and achieves performance comparable to SOTA text retrieval models on BEIR. By enhancing retrieval capabilities, we significantly improve end-to-end retrieval-augmented generation for code-related tasks. Bridging the gap between text and code retrieval, we release our models to foster research and innovation in developer tools and programming language understanding.

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

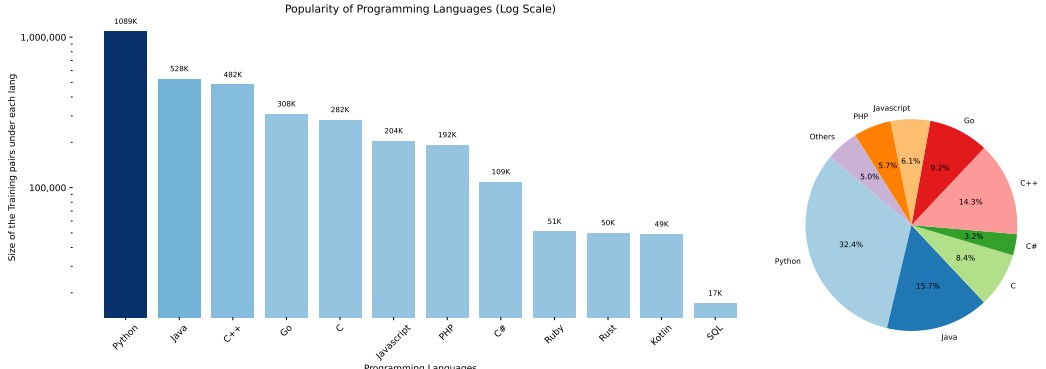

Figure 4: The programming language distribution of code training data in the general training stage.

# A   Appendix

## A.1   Dataset Details

Our training dataset contains a total of 3.36M training pairs, covering 12 different programming languages. As shown in Figure 4, the distribution of programming languages is imbalanced, with the majority of the data concentrated in a few popular languages. Python represents the largest portion of the dataset at 27.1%, followed by Go with 25.2%, and JavaScript and PHP at 17.0% and 17.2%, respectively. The remaining languages, including Java, SQL, Ruby, and others, account for smaller proportions, with Rust, Kotlin, and C# making up the smallest shares of the dataset.

**Text-to-Code Retrieval**: Text2SQL: We follow Text2SQL works in Yu et al. (2018) and Zhong et al. (2017), converting the text part into the queries to retrieval SQL as the documents, derived from Wikipedia data. In contrast, the CoIR benchmark uses synthetic-text2sql, generated using weak and strong LLMs. Code Contest: We extract the Code Contest tasks in the Khan et al. (2024), which has 11 programming language like java, c, c# et al. While CoIR benchmark use APPS, which is the interview question only in python programming language.

**Code-to-Text Retrieval**: Code Summary: We follow Hu et al. (2018) and Liu et al. (2020), which summarize code based on associated documentation, while CoIR utilizes a CodeSearchNet variant with code-comment pairs for the summary task. Code Clone Detection: We follow Svajlenko & Roy (2015) and Lu et al. (2021) to convert the code as query and its clone code as the docs ; CoIR has no dataset for this task.

**Code-to-Code Retrieval**: Code Translation: We follows Khan et al. (2024) and Lu et al. (2021), converting code in one language into queries and corresponding code in another language into documents. CoIR has no dataset for this task. Code Completion: We use the code completion tasks from Khan et al. (2024) and Lu et al. (2021), while CoIR uses a modified CodeSearchNet, splitting code snippets into query-document pairs.

**Hybrid Code Retrieval**: Code Issue Fix: We follow Khan et al. (2024) and Lu et al. (2021) to involve the code issue fix task; CoIR has no dataset for this task. Code Agent Conversation: We process the conversation data from Team et al., filtering answers based on favorite scores and excluding data used in CoIR.

## A.2   Impact of the Base Models

We analyze the base model's impact by comparing (1) text retrieval vs. generation models for training and (2) a code-specific generation model offers advantages over a general language model.

| Dataset | gte-Qwen2 Initial | gte-Qwen2 GT | Gemma-v2 GT | SFR-v2 Initial | SFR-v2 GT | Mistral GT |
|---|---|---|---|---|---|---|
| Size | 1.5B | | 2B | 7B | | 7B |
| CoIR | 62.96 | 68.52 | 67.41 | 61.28 | 69.72 | **70.40** |
| BEIR | 58.29 | 59.12 | 57.73 | 60.18 | **60.62** | 60.06 |

Table 8: Comparison of base models: Retrieval vs. Generation Models. GT represents our general training.

| Dataset | Code-Specific LLMs StarCoder-v2 3B | Code-Specific LLMs DeepSeek-Coder 6.7B | General-Domain LLMs Gemma-v2 2B | General-Domain LLMs Mistral 7B |
|---|---|---|---|---|
| CoIR | 66.95 | **71.66** | 67.41 | 70.40 |
| BEIR | 49.06 | 50.22 | 57.73 | **60.06** |

Table 9: Comparison of base models: Code-specific vs. General-domain Generation Models.

### A.2.1 Embedding Models v.s. LLMs

As shown in Table 8, the text retrieval model offers a stronger starting point, and additional training with our approach enhances both its text and code retrieval capabilities. For instance, gte-Qwen2 [10]'s CoIR performance improves from 62.96 to 68.52, while its text performance increases from 58.29 to 59.12. In contrast, the text generation model requires more extensive fine-tuning to reach similar performance. However, the advantage of text retrieval models can sometimes hinder code retrieval performance, as seen with SFR-V2 Meng* et al. (2024) underperforming compared to Mistral in specific tasks. This suggests that prior knowledge from text-focused models may not always transfer well to code-specific scenarios. To pursue a more general training approach, we chose to train using a generation model rather than a text retrieval model.

### A.2.2 Code-Specific LLMs v.s. General LLMs

We evaluate whether to choose code-specific models Lozhkov et al. (2024); Guo et al. (2024) or general LLMs Jiang et al. (2023); Team et al. (2024). As shown in Table 9, code-specific LLMs excel in code tasks but underperform in text tasks, while general LLMs perform well in both. This suggests that recent advancements in general LLMs have integrated code data into their training Team et al. (2024), and this capability can be effectively transferred to code retrieval. This finding highlights the versatility of general LLMs, making them viable for both text and code retrieval without the need for specialized models.

## A.3 Implementation Details

We summarize the base model detail in Table 10 and hyperparameters in Table 11. For the code training data, we prepend a prompt to the query in the format: "Instruct: Given Code or Text, retrieve relevant content. Query:". We use

| Model Name | Model Size | Version Number | Date of Release |
|---|---|---|---|
| gte-large-en-v1.5 | 400M | v1.5 | Approximately November 2023 |
| gemma-2-2b-it | 2B | v1 | Approximately June 2024 |
| Mistral-7B-Instruct-v0.3 | 7B | v0.3 | May 2024 |

Table 10: Model details including name, size, version, and release date.

---

[10]https://www.aimodels.fyi/models/huggingFace/gte-qwen2-15b-instruct-alibaba-nlp

| | CodeXEmbed$_{400M}$ | CodeXEmbed$_{2B}$ | CodeXEmbed$_{7B}$ |
|---|---|---|---|
| base model | gte-large-en-v1.5 | gemma-2-2b-it | Mistral-7B-Instruct-v0.3 |
| max learning rate | $5 \times 10^{-5}$ | $5 \times 10^{-5}$ | $5 \times 10^{-5}$ |
| random seed | 42 | 42 | 42 |
| GradCache | 4 | 16 | 32 |
| tuning parameters | Fully Model | LoRA | LoRA |
| Lora Rank | - | 8 | 8 |
| warmup steps | 100 | 50 | 50 |
| batch size | 1024 | 1024 | 1024 |
| max length | 512 | 512 | 512 |
| weight decay | 0.01 | 0.01 | 0.01 |
| hard negatives | 7 | 7 | 7 |
| bidirectional attention | ✓ | ✓ | ✓ |
| text:code ratio | 1:1 | 1:3 | 1:3 |
| pooling | bos | eos | eos |

Table 11: Hyperparameters for contrastive code and text training

