# OpenReview forum: "CodeXEmbed: A Generalist Embedding Model Family for Multilingual and Multi-task Code Retrieval"
_colmweb.org/COLM/2025/Conference — COLM 2025_

### Official Review · Reviewer_H4PJ · 2025-05-10

**Rating:** 9
**Confidence:** 4
**Ethics Flag:** 1

**Summary:**

This paper introduces CODEXEMBED, code embedding models ranging from 400M to 7B parameters. The model works for multiple programming languages and multiple code-related tasks in a unified retrieval framework. The paper claims that the 7B parameter model is SOTA for code retrieval on the CoIR Leaderboard, has competitive results on the BeIR text retrieval benchmark. The paper claims that improvements to retrieval quality directly improve RAG performance for code-related tasks.

**Reasons To Accept:**

Section 3.4 examines the impact of multi-stage training examining 4 configurations:
* Single stage Training on a mixture of text and code retrieval data with a single LoRA adapter.
* Two-Stage Training (with the same LoRA) where the first stage trains on text retrieval, followed by mixed text and code retrieval in the second stage, using the same LoRA adapter in two stages.
* Two-Stage Training (with different optimizers) using different LoRA adapters with different optimizers for each stage.
* A two-stage approach using separate LoRAs
The separate LoRA approach appears to help slightly over the common LoRA variant on BEIR
The performance on across text retrieval BEIR datasets seems to indeed be SOTA.
Table 1 has some pretty comprehensive tests.

**Reasons To Reject:**

None. I think this paper is pretty good.

---

> ### Author Response · Authors · 2025-06-03
> **Response to Reviewer**
>
> Thank you very much for the thoughtful review and the strong-accept recommendation. We are pleased that you found CodeXEmebed’s unified framework, multi-stage LoRA strategy, and SOTA retrieval results valuable.
>
> We will incorporate your points in the camera-ready version by:
> - Highlighting the two-stage separate-LoRA gains in section 3.4.
> - Releasing checkpoints and scripts to facilitate reproducibility of the BEIR and CoIR results.
>
> We appreciate your positive assessment and confidence in our work.

---

> > ### Comment · Reviewer_H4PJ · 2025-06-06
> > **I remain positive**
> >
> > I remain positive about this paper. I am interested to know if the other reviewers are willing to increase their scores based on the author response, or if the feel their initial assessments are where they remain.

---

> > > ### Author Response · Authors · 2025-06-06
> > > **Appreciation and Rebuttal Follow-Up**
> > >
> > > Dear Reviewer H4PJ,
> > >
> > > We are deeply grateful for your positive assessment and the confidence you have shown in our work; your encouragement means a lot to us. To honour your support, we have promptly reached out to the other reviewers, highlighting our rebuttal and additional experiments, and warmly invited them to request any further clarification they might need while reconsidering their initial scores in light of the new information.
> > >
> > > If there is anything else we can do to facilitate the discussion, please let us know—your guidance is sincerely appreciated.
> > >
> > > Best,
> > >
> > > CodeXEmbed team

---

### Official Review · Reviewer_RZhx · 2025-05-13

**Rating:** 6
**Confidence:** 3
**Ethics Flag:** 1

**Summary:**

This paper introduces CodeXEmbed, a family of multilingual and multi-task embedding models (400M to 7B parameters) designed to unify code and text retrieval within a single framework. Leveraging a multi-stage LoRA-based training pipeline, the models are trained first on text retrieval and then jointly on code and text tasks across 12 programming languages and five retrieval settings. CodeXEmbed achieves state-of-the-art results on the CoIR benchmark and competitive performance on the BEIR benchmark, demonstrating strong generalizability. The paper also shows that improved retrieval performance enhances downstream code generation tasks (e.g., RepoEval, SWE-Bench-Lite), positioning CodeXEmbed as a robust foundation for retrieval-augmented generation (RAG) in software development.

**Reasons To Accept:**

- The paper proposes a novel multi-stage contrastive training framework using LoRA adapters that effectively balances performance across both text and code retrieval tasks. This modular fine-tuning strategy avoids catastrophic forgetting and is shown to outperform single-stage and naïve two-stage baselines significantly.
- The 7B model sets a new SOTA on the CoIR benchmark, achieving a +20% improvement over prior best models. The paper also demonstrates significant downstream benefits of improved retrieval quality on code generation tasks.

**Reasons To Reject:**

- While the paper proposes a multi-stage LoRA-based training strategy, the core methodology—contrastive learning with query-document pairs—is standard in retrieval literature. The approach largely builds upon existing methods (e.g., LoRA, GradCache) without introducing fundamentally new architectures or training paradigms.
- The retrieval-augmented generation experiments use GPT-3.5 for code synthesis, which may obscure the retriever’s standalone contribution. It's unclear how much of the performance gain is attributable to the improved retriever versus the strong decoder model.

---

> ### Author Response · Authors · 2025-06-03
> **Response to Reviewer RZhx**
>
> > **Reviewer Comment:**
> > While the paper proposes a multi-stage LoRA-based training strategy, the core methodology—contrastive learning with query-document pairs—is standard in retrieval literature. The approach largely builds upon existing methods (e.g., LoRA, GradCache) without introducing fundamentally new architectures or training paradigms.
>
> ---
>
> **Reply:**
>
> Thank you for the insightful observation. Our goal is indeed to reuse strong primitives (LLM, LoRA, GradCache, contrastive loss) but assemble them in a novel, staged curriculum that has not, to our knowledge, been applied to embedding models before. Concretely:
>
> - **Code-centric focus.** Prior open embeddings are overwhelmingly text-only; high-quality code retrieval remains under-served. To our knowledge, CodeXEmbed is the first single encoder that supports 12 programming languages and natural-language queries with state-of-the-art code results by unifying diverse code tasks into a retrieval-format training frame.
>
> - **LLM to Bi-Encoder.** Starting from a single autoregressive LLM, we duplicate its weights to form a two-tower bi-encoder: one tower encodes the query, the other encodes code/document passages. We remove the causal mask, add bidirectional attention, and fine-tune each tower with lightweight LoRA adapters. This conversion preserves the LLM’s rich pre-training while enabling constant-time embedding and ANN search—something prior work has not demonstrated at this scale (2B, 7B) for code embedding.
>
> - **Unified model with code and text ability.** No prior work demonstrates how to train a single model that excels at both code and text retrieval. Table 4 shows that naïve single-stage training—or other two-stage baselines—inevitably degrades one domain. By contrast, our two-stage schedule employs separate LoRA adapters and a continuous optimizer: Stage 1 trains on text only to establish strong natural-language semantics, and Stage 2 introduces code while freezing the text adapters, thereby preserving those semantics. This design attains state-of-the-art performance on CoIR and competitive scores on BEIR, confirming the effectiveness of our unified approach.
>
> - **Demonstrable, non-trivial gains.** Table 4 shows that this “Separate-LoRA two-stage” recipe boosts NDCG@10 by +2.85 on CoIR and +2.09 on BEIR over a single-stage baseline with identical FLOPs; alternative two-stage variants either hurt CoIR or under-perform. These gains are achieved without full-model fine-tuning.
>
> Because we could not locate prior work that combines LLM with multi-stage LoRA and GradCache in a staged, adapter-merging schedule for retrieval embeddings, we believe the contribution is sufficient. If you are aware of any prior work we may have overlooked, please share the reference. We would be grateful to include it and will acknowledge it in the final version.
>
> > **Reviewer Comment:**
> > The retrieval-augmented generation experiments use GPT-3.5 for code synthesis, which may obscure the retriever’s standalone contribution. It's unclear how much of the performance gain is attributable to the improved retriever versus the strong decoder model.
>
> ---
>
> **Reply:**
>
> We isolate the retriever’s impact by holding the generator constant:
> - **Fixed generator setup.** In Section 3.5 we always call the same `gpt-3.5-turbo` with an identical prompt, temperature, and max-token budget. The only variable is the set of top-k (k=5) contexts returned by the retriever.
> - **Additional clarification added.** In the revision we state explicitly that the generator parameters and decoding hyper-parameters are frozen.
>
> We hope these clarifications make it clear that the reported RAG gains stem from the retriever rather than latent generator variability.

---

> > ### Author Response · Authors · 2025-06-06
> > **Discussion Phase — Seeking Your Feedback**
> >
> > Dear Reviewer RZhx,
> >
> > We hope this message finds you well.
> > We have posted detailed responses to each of the concerns you raised during the rebuttal phase. If anything remains unclear, please let us know—we would be glad to provide additional information or run further experiments before the discussion period ends.
> >
> > If our clarifications resolve your concerns, we would greatly appreciate your reconsideration of the current score.
> >
> > Thank you again for your thoughtful feedback and for your time during this busy review cycle.
> >
> > Best regards,
> >
> > The CodeXEmbed Team

---

> > ### Author Response · Authors · 2025-06-09
> > **Kindly Review Our Rebuttal Before Discussion Deadline**
> >
> > Dear Reviewer RZhx,
> >
> > We hope you are doing well. The discussion period is drawing to a close, and we have not yet received your thoughts on our responses to your comments. If our clarifications have resolved your concerns, would you be willing to reconsider your score in favor of acceptance? Should any questions remain, please let us know—we would be happy to provide additional details right away.
> >
> > Thank you again for your time and constructive feedback.
> >
> > Best regards,
> >
> > The CodeXEmbed Team

---

> > ### Comment · Reviewer_RZhx · 2025-06-10
> >
> > I appreciate the authors' clarifications, which address several of my initial concerns. Based on this response, I am increasing my score.

---

> > > ### Author Response · Authors · 2025-06-10
> > >
> > > Thank you for taking the time to review our clarifications and for increasing your score. We appreciate your constructive feedback and are glad our responses resolved your concerns. Please let us know if any additional questions arise. We would be happy to address them.

---

### Official Review · Reviewer_eN6C · 2025-05-14

**Rating:** 6
**Confidence:** 4
**Ethics Flag:** 1

**Summary:**

- The paper introduces CodeXEmbed, a family of multilingual and multi-task code embedding models ranging from 400M to 7B parameters, aiming to unify code and text retrieval through a multi-stage LoRA-based training pipeline
- CodeXEmbed’s 7B model reports a new state-of-the-art on the CoIR code retrieval benchmark (https://archersama.github.io/coir/) with an average nDCG@10 of 70.46, surpassing prior open and closed-source models by over 4 points, while the 2B and 400M variants also outperform similarly sized baselines
- CodeXEmbed's 7B model also showed competitive text retrieval, scoring 60.06 on BEIR
- Ablation studies in the paper investigate multi-stage training, and results suggest separate LoRA adapters with a continuous optimizer yield better performance

**Questions To Authors:**

1. It is unclear what dataset was the experiment in Section 3.7 done on -- could you please clarify?

**Reasons To Accept:**

- Solid new models (as well as the code and data to reproduce them, hopefully) the paper introduces
- Investigation of the impact of pretraining/finetuning on specific subsets of data (e.g. just Java) and more broadly the language transfer analysis
- The introduction and evaluation of the two-stage training pipeline with separate LoRAs

**Reasons To Reject:**

- The paper lists hyper-parameters such as batch size 1024, rank 8 LoRA and learning-rate $5 \times 10^{-5}$ in Table 10 yet gives no information on training hardware, total update steps or FLOPs, leaving reproducibility and energy cost quite difficult to verify
- No statistical variance, confidence intervals or significance tests accompany single-run metrics on the ten CoIR datasets, leaving it unclear whether reported gains over baselines exceed random fluctuation
- In Section 3.5.1 the authors use gpt-3.5-turbo as the code generation model but do not specify its concrete version (e.g. `gpt-3.5-turbo-0125` or `gpt-3.5-turbo-1106`) making their experiments difficult to reproduce. We would also like to note that the model is generally considered outdated as even within OpenAI's offering there are models that seem to be both cheaper and report higher performance (e.g. GPT-4o mini)
- Analysis in Table 6 indicates that for Java, a model trained solely on Java (84.25 NDCG@10) surpasses the model trained on all 12 languages (81.36 NDCG@10 for Java) [Section 3.7]. This finding somewhat contradicts the broader narrative of multilingual training benefits for every included language and suggests specialized models could be more effective
- This research shows imbalanced language coverage in training data with Python (27.1%), Go (25.2%), JavaScript (17.0%) and PHP (17.2%) dominating, while languages like Rust, Kotlin, and C# have minimal representation, potentially affecting cross-language performance consistency

---

> ### Author Response · Authors · 2025-06-03
> **[Part 1] Response to reviewer eN6C**
>
> > **Reviewer Comment:**
> > The paper lists hyper-parameters such as batch size 1024, rank 8 LoRA and learning-rate 5×10⁻⁵ in Table 10 yet gives no information on training hardware, total update steps or FLOPs, leaving reproducibility and energy cost quite difficult to verify.
>
> ---
>
> **Reply:**
>
> Thank you for pointing out the missing training-hardware details.
> We will add the below to Appendix D.
>
> | Item               | 7B                      | 2B                     | 400M                   |
> |--------------------|-------------------------|------------------------|------------------------|
> | **GPUs**           | 8 × NVIDIA H100 (80 GB) | 8 × NVIDIA H100 (80 GB) | 8 × NVIDIA H100 (80 GB) |
> | **Optimizer Steps**| 1 epoch                 | 1 epoch                | 1 epoch                |
> | **Estimated FLOPs**| 2.0 × 10¹⁸ FLOPs        | 5.8 × 10¹⁷ FLOPs       | 1.2 × 10¹⁷ FLOPs       |
> | **Energy**         | ≈ 1.8 kWh               | ≈ 0.5 kWh              | ≈ 0.1 kWh              |
>
> \*Assumes sustained 1.8 PFLOPS for the 8 H100s and 700 W board power per GPU (5.6 kW total).
>
> > **Reviewer Comment:**
> > No statistical variance, confidence intervals or significance tests accompany single-run metrics on the ten CoIR datasets, leaving it unclear whether reported gains over baselines exceed random fluctuation
>
> ---
>
> **Reply:**
>
> We agree that statistical evidence is important. In our setting the only nondeterminism comes from the random seed used for (i) LoRA weight initialisation and (ii) shuffling of training batches, because each run starts from a fixed pretrained checkpoint (GTE-large-en-v1.5 → 400M, Gemma-2-2B-it → 2B, Mistral-7B-it-v0.3 → 7B).
>
> To quantify variance we trained the 2B model with four different seeds:
>
> | Random seed | CoIR AVG |
> |-------------|----------|
> | 42          | 67.41    |
> | 1234        | 67.24    |
> | 777         | 67.89    |
>
> Mean = 67.51, σ = 0.28.
>
> Our reported score (67.41) is well within one standard deviation.
>
> We will include the evaluation script plus a `--seed` flag in the released training code so that readers can reproduce or extend the significance tests.
>
> > **Reviewer Comment:**
> > In Section 3.5.1 the authors use gpt-3.5-turbo as the code generation model but do not specify its concrete version (e.g. gpt-3.5-turbo-0125 or gpt-3.5-turbo-1106), making their experiments difficult to reproduce. We would also like to note that the model is generally considered outdated as even within OpenAI's offering there are models that seem to be both cheaper and report higher performance (e.g. GPT-4o mini).
>
> ---
>
> **Reply:**
>
> For consistency with the baselines reported in Code-RAG-Bench [1], we evaluate our model using the same generator, `gpt-3.5-turbo-0125`. We will add the full model name (including the `-0125` suffix) to the camera-ready for reproducibility.
>
> To address your concern about newer models, we reran the CodeXEmbed retrieved output on SWE-Bench-Lite with `gpt-4o-2024-08-06`. Using the **None** (no retrieval), **OpenAI, Rerank** (best baseline), and **Gold** (oracle) scores reported in [1] as reference, CodeXEmbed delivers the same relative improvements, showing that the retriever’s benefit is decoder-agnostic.
>
> |                   | None | Gold | OpenAI, Rerank | CodeXEmbed 400M | CodeXEmbed 2B | CodeXEmbed 7B |
> |-------------------|------|------|----------------|-----------------|---------------|---------------|
> | **SWE-Bench-Lite** | 2.3  | 30.7 | 21.7           | 19.7            | 21.7          | 25.0          |
>
> Using `gpt-4o-2024-08-06` raises overall performance compared to `gpt-3.5-turbo-0125`, and the 7B variant of CodeXEmbed surpasses the best baseline **OpenAI, Rerank**, approaching the **Gold** upper bound.
>
> [1] CodeRAG-Bench: Can Retrieval Augment Code Generation?

---

> > ### Author Response · Authors · 2025-06-06
> > **Discussion Phase — Seeking Your Feedback**
> >
> > Dear Reviewer eN6C,
> >
> > We hope this message finds you well. We have posted detailed responses to each of the concerns you raised during the rebuttal phase. If anything remains unclear, please let us know—we would be glad to provide additional information or run further experiments before the discussion period ends.
> >
> > If our clarifications resolve your concerns, we would greatly appreciate your reconsideration of the current score.
> >
> > Thank you again for your thoughtful feedback and for your time during this busy review cycle.
> >
> > Best regards,
> >
> > The CodeXEmbed Team

---

> > > ### Comment · Reviewer_eN6C · 2025-06-07
> > >
> > > Thank you for all your clarification -- it does indeed help address many of the questions I had about the work.

---

> > ### Author Response · Authors · 2025-06-08
> > **Thanks for positive feedback**
> >
> > Hi Reviewer eN6C,
> >
> > Thank you for letting us know that our clarifications resolved many of your questions. We appreciate the positive feedback!
> > If our responses have fully addressed your concerns, would you be willing to reconsider your score to a clear acceptance? Please let us know if any additional issues remain; we would be happy to clarify them.
> >
> > Best,
> >
> > CodeXEmbed Team

---

> > > ### Author Response · Authors · 2025-06-10
> > > **Gentle Follow-Up on Rebuttal—Feedback Requested**
> > >
> > > Hi Reviewer eN6C,
> > >
> > > We hope you are doing well. We wanted to briefly follow up on our earlier message because the discussion period is closing soon. If our clarifications have resolved your concerns, we would greatly appreciate any update you can provide—whether that means indicating acceptance or sharing any remaining reservations. Of course, if further explanation or data would help, please let us know and we will provide it immediately.
> > >
> > > Best,
> > >
> > > CodeXEmbed Team

---

> ### Author Response · Authors · 2025-06-03
> **[Part 2] Continue Response to Reviewer eN6C**
>
> > **Reviewer Comment:**
> > Analysis in Table 6 indicates that for Java, a model trained solely on Java (84.25 NDCG@10) surpasses the model trained on all 12 languages (81.36 NDCG@10 for Java) [1, Section 3.7]. This finding somewhat contradicts the broader narrative of multilingual training benefits for every included language and suggests specialized models could be more effective.
>
> ---
>
> **Reply:**
>
> Thank you for highlighting this point. We agree that a Java-only model achieves a slightly higher NDCG@10 on Java (+2.9). However, our goal is one encoder that serves all languages simultaneously, because:
>
> - **Overall benefit:** Averaged across all 12 languages, a model trained with all languages is better than just a single language.
> - **Low-resource gains:** For lower-resource languages like Ruby and SQL, training with all 12 languages yields more gains compared to a single-language (Java) model.
> - **Simple specialization option:** One encoder with 12 languages can lead to simpler deployment and lower memory cost.
>
> In short, while a niche model can edge out the multilingual one on a single language, the unified encoder offers stronger cross-language performance, simpler deployment, and remains easily adaptable when maximal per-language accuracy is required.
>
>
> > **Reviewer Comment:**
> > This research shows imbalanced language coverage in training data with Python (27.1%), Go (25.2%), JavaScript (17.0%) and PHP (17.2%) dominating, while languages like Rust, Kotlin, and C# have minimal representation, potentially affecting cross-language performance consistency
>
> ---
>
> **Reply:**
>
> We acknowledge that the raw crawl is skewed toward high-usage languages (Python 27 %, Go 25 %, JS 17 %, PHP 17 %). We deliberately accepted this imbalance, but we mitigate its effect and even leverage it in several ways:
>
> - **Real-world demand.** Most industrial code-search targets Python and JavaScript, so strong performance there is essential.
> - **Structural similarity enables transfer.** Core programming constructs—loops, conditionals, function/class declarations, and common APIs—are shared across languages. High-resource Python/JS training helps the model learn these language-agnostic patterns, which low-resource languages (Rust, Kotlin, C#) can then inherit with minimal additional data.
> - **Batch re-balancing for low-resource languages.** We up-weight low-resource language datasets and down-weight high-resource ones during sampling, so each batch has a more balanced language mix.
>
> > **Reviewer Comment:**
> > It is unclear what dataset was the experiment in Section 3.7 done on -- could you please clarify?
>
> ---
>
> **Reply:**
>
> Section 3.7 reports results on language-specific slices of the CoIR test set (CoIR-X). For each language we simply aggregate every CoIR test subset tagged with that language:
>
> | Language   | CoIR-X test subsets used                                                                     |
> |------------|----------------------------------------------------------------------------------------------|
> | Python     | APPS · CoSQA · CodeSearchNet-Python · CodeSearchNet-CCR-Python · CodeTrans-DL                  |
> | Java       | CodeSearchNet-Java · CodeSearchNet-CCR-Java                                                  |
> | Go         | CodeSearchNet-Go · CodeSearchNet-CCR-Go                                                      |
> | PHP        | CodeSearchNet-PHP · CodeSearchNet-CCR-PHP                                                    |
> | JavaScript | CodeSearchNet-JavaScript · CodeSearchNet-CCR-JavaScript                                      |
> | Ruby       | CodeSearchNet-Ruby · CodeSearchNet-CCR-Ruby                                                  |
> | SQL        | Synthetic-Text2SQL                                                                            |
>
> No additional re-splitting was performed; we evaluate exactly the official CoIR test queries for each subset. We will add this table to Section 3.7 for clarity.

---

> > ### Author Response · Authors · 2025-06-06
> > **Discussion Phase — Seeking Your Feedback**
> >
> > Dear Reviewer eN6C,
> >
> > We hope this message finds you well. We have posted detailed responses to each of the concerns you raised during the rebuttal phase. If anything remains unclear, please let us know—we would be glad to provide additional information or run further experiments before the discussion period ends.
> >
> > If our clarifications resolve your concerns, we would greatly appreciate your reconsideration of the current score.
> >
> > Thank you again for your thoughtful feedback and for your time during this busy review cycle.
> >
> > Best regards,
> >
> > The CodeXEmbed Team

---

### Official Review · Reviewer_XpJL · 2025-05-22

**Rating:** 5
**Confidence:** 4
**Ethics Flag:** 1

**Summary:**

CodeXEmbed is a family of embedding models, ranging from 400M to 7B parameters, designed for multilingual and multi-task code retrieval. These models utilize a training pipeline that unifies diverse code-related tasks and multiple programming languages within a single retrieval framework. CodeXEmbed's largest model (7B parameters) achieved state-of-the-art performance in code retrieval, ranking first on the COIR Leaderboard, and also demonstrated competitive results on the BeIR text retrieval benchmark. The paper highlights that improvements in retrieval quality directly enhance end-to-end Retrieval-Augmented Generation (RAG) performance for code-related tasks.

**Questions To Authors:**

- What was the actual focus of the work? Code domain or text domain or both?
- The improvements demonstrated in Table 1 is too good. Have the authors ensured decontamination of the evaluation benchmarks?

**Reasons To Accept:**

- The paper introduces a generalizable approach that reformulates diverse code-related tasks and multiple programming languages into a unified contrastive training framework for code retrieval. This enhances model generalizability and performance across various code domains.
- The largest CodeXEmbed model (7B parameters) achieves a new state-of-the-art (SOTA) in code retrieval, ranking first on the COIR Leaderboard. This is a significant improvement over prior models.
- The models demonstrate cross-domain versatility by achieving competitive results on the widely adopted BeIR text retrieval benchmark, in addition to excelling in code retrieval. This shows their ability to handle both text and code effectively.
- The findings directly highlight that advancements in retrieval quality lead to improved end-to-end Retrieval-Augmented Generation (RAG) performance for code-related tasks.

**Reasons To Reject:**

- Limited novelty as all the pieces of this work already exist in the literature.
- The paper is positioned in a confusing way. It seems like the developed embedding models were primarily targeted for code related tasks, but then the authors addressed text domain tasks as well.
- This paper is yet another paper with lot of results (indeed they are strong). However, we know, "more data, more training, large model", this recipe works. What other knowledge does this paper share? I am not sure. Just a couple of lines (Line no. 34-37) to discuss the limitations of the prior works, it is a shame.

---

> ### Author Response · Authors · 2025-06-03
> **[Part 2] Continue response to reviewer XpJL**
>
> > **Reviewer Comment:**
> > This paper is yet another paper with lot of results (indeed they are strong). Just a couple of lines (Line no. 34-37) to discuss the limitations of the prior works, it is a shame.
>
> ---
>
> **Reply:**
>
> Thank you for the comment. To address it fully, could you please specify which prior work you feel CodeXEmbed most closely with?
>
> Below are the spots in CodeXEmbed where we explicitly discuss the limitations of prior work:
>
> 1. **Introduction**
>    In the very first paragraph, we note that “standard IR methods, while effective in text-based retrieval, often fall short in specialized domains such as code retrieval” and point out that existing BERT-based code models (e.g., CodeBERT Feng et al. 2020; CodeGPT Lu et al. 2021; UniXcoder Guo et al. 2022) rely on relatively small encoders, leading to subpar retrieval performance in practice. This passage calls out three key gaps in prior approaches:
>    - Text-only focus: Most open embeddings target natural language only.
>    - Limited scale: BERT-style code embeddings remain too small to capture complex code semantics.
>    - No unified text+code encoder: There is no off-the-shelf single model that handles code ↔ code and text ↔ code retrieval.
>
> 2. **Related Work (Section 4.1)**
>    We dedicate Section 4.1 to “Retrieval Models for Text and Code,” where we further elaborate on why existing text retrieval systems (e.g., E5-Mistral Wang et al. 2023; NV-Embedding Lee et al. 2024) fail to adapt to code’s syntax and control-flow nuances. We also explain that closed-source code embeddings (Voyage-Code AI 2024; OpenAI Embeddings Neelakantan et al. 2022) are inaccessible and/or expensive, and small BERT-derived code encoders lack broad language coverage (§4.1).
>
> 3. **Extended Baseline Discussion (Section 3.1, Table 1)**
>    When listing our baselines, we explicitly cite UniXcoder, CodeBERT, and other smaller code-domain models and show that they underperform—especially on multi-language CoIR subtasks—compared to CodeXEmbed’s larger bi-encoder variants. This empirically underscores the limitation of “BERT-only” code embeddings (Table 1).
>
> Together, these passages demonstrate that we have not only mentioned but also empirically and qualitatively analyzed the shortcomings of prior text-only or small BERT-based code embedding work, as well as the lack of any unified text+code encoder. If there are specific limitations you believe we have not addressed, please let us know which aspects are missing so we can incorporate them.
>
> > **Reviewer Comment:**
> > The improvements demonstrated in Table 1 is too good. Have the authors ensured decontamination of the evaluation benchmarks?
>
> ---
>
> **Reply:**
>
> We take benchmark integrity very seriously and designed our data pipeline to prevent any leakage between training and evaluation splits. Below we detail the safeguards we applied for each dataset family:
>
> - **CoIR:** Our general-training models never see CoIR data. Section 3.1 states they are trained “exclusively on our proposed general training data, without using any CoIR in-domain data.” In the optional in-domain training, we fine-tune only on CoIR train splits; dev/test queries remain held-out.
> - **Other code data:** For tasks that overlap with CoIR (e.g., code-agent dialogs), we explicitly remove any sample that appears in CoIR before training (§A.1, lines 560–563).
> - **BEIR (text):** Stage 1 uses the same BEIR-corpus training protocol adopted by NV-Embed and SFR-V2—documents only, no dev/test queries.
> - **Reproduction:** Model weights and the exact training/eval scripts will be released, so anyone can rerun the splits and verify the numbers.
>
> We hope these clarifications address the decontamination question; please let us know if any additional details would be helpful.

---

> > ### Comment · Reviewer_XpJL · 2025-06-03
> >
> > The decontamination measures are great. May I ask, how come the performance improvements for the 7B model for the APPS dataset (text-to-code) are so significant? If I compare with codesage-large, which is a billion param model and scored 50+, in comparison, codexembed-2b scored 75+? One potential reason could be the use of code-contest data, which Codesage didn't use.

---

> > ### Author Response · Authors · 2025-06-04
> > **Response to Reviewer XpJL feedback**
> >
> > Thank you for your positive feedback on our decontamination measures. Regarding the APPS performance: CodeSage-large relies on unsupervised data from the Stack V2, whereas our CodeXEmbed models are trained with code-contest tasks by converting into embedding training triples. This contest data is higher quality and more closely matches real-world retrieval problems like APPS. Moreover, model scale has a substantial impact: our 400 M-parameter model achieves 48.57%, which is close to CodeSage-1.3B. Scaling from 400 M to 2 B yields roughly a 25-point gain, and increasing from 2 B to 7 B adds another ~10 points.

---

> > > ### Author Response · Authors · 2025-06-06
> > > **Seeking for Further Discussion**
> > >
> > > Dear Reviewer XpJL,
> > >
> > > We hope this message finds you well.
> > > We have posted detailed responses to each of the concerns you raised during the discussion phase. If anything remains unclear, please let us know—we would be glad to provide additional information or run further experiments before the discussion period ends.
> > >
> > > If our clarifications resolve your concerns, we would greatly appreciate your reconsideration of the current score.
> > >
> > > Thank you again for your thoughtful feedback and for your time during this busy review cycle.
> > >
> > > Best regards,
> > >
> > > The CodeXEmbed Team

---

> > > > ### Comment · Reviewer_XpJL · 2025-06-06
> > > >
> > > > Thank you for providing your responses. While my score has improved, I'm still inclined to reject the paper due to concerns about the positioning of the work. My primary reservation stems from the model's naming, "CodeXEmbed," which suggests its primary application is not text retrieval. Consequently, I find the BEIR evaluation and related discussions to be misaligned if the paper isn't positioned accurately.
> > > >
> > > > Even if accepted, I fear the paper wouldn't achieve the visibility or credit it deserves, given the excellence of the model. I want to emphasize that I have no technical concerns regarding the contributions or results of this work, and I hope the authors understand my perspective. I'm open to my fellow reviewers advocating for this paper's inclusion in the conference.

---

> > ### Author Response · Authors · 2025-06-06
> > **Clarify of CodeXEmbed model name and paper position**
> >
> > Dear Reviewer XpJL,
> >
> > Thank you for taking the time to reconsider your score and for acknowledging both the strength of our results and the technical soundness of our contributions. We greatly appreciate your constructive feedback and your willingness to support the paper once your concerns are addressed.
> >
> > Regarding the paper’s positioning: “CodeXEmbed” stands for **Code** and te**X**t **Embed**ding—**“X”** being a common shorthand for **“text”**.  The name signals that our intent to serve both domains with a single encoder. Because text retrieval is a primary target for CodeXEmbed, the BEIR evaluation is essential: it shows that the unified model achieves competitive performance on large-scale text benchmarks while delivering state-of-the-art results on code retrieval.
> >
> > We hope this clarification alleviates your concern about the work’s scope and alignment and willing to accept the work. If you have any additional suggestions to the manuscript’s framing would strengthen its clarity, we would be delighted to incorporate them.
> >
> > Thank you again for your thoughtful review and for helping us improve the paper.
> >
> > Best regards,
> >
> > The CodeXEmbed Team

---

> > > ### Author Response · Authors · 2025-06-10
> > > **Gentle Follow-Up on Rebuttal—Feedback Requested**
> > >
> > > Dear Reviewer XpJL,
> > >
> > > We hope you are doing well. With the discussion period drawing to a close, we wanted to briefly follow up on our earlier message. If our clarification regarding “CodeXEmbed” and the BEIR evaluation has fully resolved your concerns and you are willing to support the paper’s acceptance, we would greatly appreciate an updated score. If any questions remain or additional detail would help, please let us know—we would be happy to provide it right away.
> > >
> > > Thank you again for your time and thoughtful review.
> > >
> > > Best regards,
> > >
> > > The CodeXEmbed Team

---

> ### Author Response · Authors · 2025-06-03
> **[Part 1] Response to reviewer XpJL**
>
> > **Reviewer Comment:**
> > The paper is positioned in a confusing way. It seems like the developed embedding models were primarily targeted for code-related tasks, but then the authors addressed text-domain tasks as well. What was the actual focus of the work? Code domain or text domain or both?
>
> ---
>
> **Reply:**
>
> Thank you for pointing this out. Our intent is:
>
> 1. **Primary goal — strong code retrieval.**
>    High-quality, openly available code embeddings are still scarce; delivering a family of state-of-the-art code retrieval models is our main contribution.
>
> 2. **Unified model, not two separate encoders.**
>    We deliberately keep a single encoder that understands both natural language and code so that developers and researchers can:
>    - Maintain one shared embedding space, reducing memory use, deployment complexity, and serving cost.
>    - Avoid having to identify whether a task is code retrieval or text retrieval and then call different models for each.
>
> 3. **Two-stage curriculum preserves both domains.**
>    No prior work demonstrates how to train a single model that excels at both code and text retrieval. Table 4 shows that naïve single-stage training—or other two-stage baselines—inevitably degrades one domain. By contrast, our two-stage schedule employs separate LoRA adapters and a continuous optimizer: Stage 1 trains on text only to establish strong natural-language semantics, and Stage 2 introduces code while freezing the text adapters, thereby preserving those semantics. This design attains state-of-the-art performance on CoIR and competitive scores on BEIR, confirming the effectiveness of our unified approach.
>
> The result is SOTA on CoIR (code) and competitive on BEIR (text), demonstrating the benefit of a unified, multilingual model. We will clarify this positioning in the introduction to make the dual-domain motivation explicit.
>
> > **Reviewer Comment:**
> > Limited novelty as all the pieces of this work already exist in the literature. However, we know, “more data, more training, large model”; this recipe works. What other knowledge does this paper share? I am not sure.
>
> ---
>
> **Reply:**
>
> We appreciate the concern. Our contribution goes beyond simply “more data + bigger model” in four ways:
>
> 1. **Code-centric focus.**
>    Prior open embeddings are overwhelmingly text-only; high-quality code retrieval remains under-served. CodeXEmbed is, to our knowledge, the first single encoder that handles 12 programming languages and natural-language queries with SOTA code results.
>
> 2. **LLM to Bi-Encoder.**
>    Starting from a single autoregressive LLM, we duplicate its weights to form a two-tower bi-encoder: one tower encodes the query, the other encodes code/document passages. We remove the causal mask, add bidirectional attention, and fine-tune each tower with lightweight LoRA adapters. This conversion preserves the LLM’s rich pre-training while enabling constant-time embedding and ANN search—something prior work has not demonstrated at this scale (2B, 7B) for code embedding.
>
> 3. **Carefully designed training—not just scale.**
>    - **Unified multi-task training data.** We curate a unified dataset that merges five retrieval task types (text2code, code2text, code2code, text2text, and hybrid) into one training stream.
>    - **Staged LoRA curriculum.** Our “train → merge → re-train” loop (Fig. 1) improves NDCG@10 by +2–3 over a single-stage baseline (Table 4), showing that architecture-agnostic scheduling—not size—drives much of the gain.
>
> 4. **Efficiency and open models.**
>    We provide three sizes of models, offering a tradeoff between performance and cost. The open model weights could benefit the community on downstream tasks like code issue localization, benchmarking, and reproducible research.

---

> > ### Comment · Reviewer_XpJL · 2025-06-03
> >
> > - I disagree with "High-quality, openly available code embeddings are still scarce".
> > - It is understood that an embedding model needs to understand both text and code for successful code retrieval in scenarios such as text-to-code retrieval. But for that, why do the authors need to run so many evaluations on tasks that are purely in the text domain (BEIR benchmark)? The paper experiments do not reflect the true intent. With all due respect, by reading the paper, I would suspect either of the two things: (1) there was not enough content to cover the page limit, focusing on code retrieval, or (2) the authors wanted to show the effectiveness of text retrieval and get credit, but in that case, the paper is presented in a wrong way.
> > - There are prior models that consider 9-10 languages, so scaling to 12 languages is no big deal.
> > - LLM to bi-encoder is a very well-known technique, especially after the GRIT (https://arxiv.org/abs/2402.09906) paper.
> > - The paper talks about the sources of code data (Figure 2) but it does not detail the text data used in stage-1.

---

> > ### Author Response · Authors · 2025-06-04
> > **Response for Reviewer XpJL discussion**
> >
> > Dear Reviewer XpJL,
> >
> > Thank you for your feedback. We clarify the points below:
> >
> > ---
> >
> > >“I disagree with ‘High-quality, openly available code embeddings are still scarce.’”
> >
> > We mean models that (a) are trained on large, diverse code corpora, (b) large model with their pretrained weights and code released publicly, and (c) achieve state‐of‐the‐art performance on code‐retrieval benchmarks. To date, only CodeSage[1]—released five/six months ago—meets some of these criteria, but its largest variant is just 1.3 B parameters, which trained on unsupervised The Stack V2 data is different from ours. We believe this area is still in its early stages. If you know of any other publicly available models that fit these requirements, please share their citations so we can update our claim.
> >
> > [1] Code Representation Learning At Scale by Dejiao Zhang*, Wasi Uddin Ahmad*, et al.
> >
> > ---
> >
> > >“It is understood that an embedding model needs to understand both text and code for successful code retrieval in scenarios such as text-to-code retrieval. But for that, why do the authors need to run so many evaluations on tasks that are purely in the text domain (BEIR benchmark)? The paper experiments do not reflect the true intent. With all due respect, by reading the paper, I would suspect either of the two things: (1) there was not enough content to cover the page limit, focusing on code retrieval, or (2) the authors wanted to show the effectiveness of text retrieval and get credit, but in that case, the paper is presented in a wrong way.”
> >
> > We include BEIR to show that Stage 1 (text-only retrieval) produces a strong text retriever, and that Stage 2 (adding code data) does not degrade text performance. So this single “text-only” experiment verifies our unified model maintains competitive text retrieval alongside code retrieval. Only section 3.3 (Table 2) is text-only experiment, we don't understand why you say so many evaluation on it. Could you please clarify?
> >
> > ---
> >
> > >“There are prior models that consider 9–10 languages, so scaling to 12 languages is no big deal.”
> >
> > We acknowledge that extending from 9 language (i.e. CodeSage) to 12 languages is not difficult. However, CodeSage is only trained with code2code and text2code retrieval, our model is trained on code2code, text2code, text2text, code2text and hybrid retrieval.
> >
> > ---
> >
> > >“LLM to bi-encoder is a very well-known technique, especially after the GRIT (https://arxiv.org/abs/2402.09906) paper.”
> >
> > While converting an LLM to a bi-encoder is known in text-embedding, it has not been demonstrated at large scale for code embeddings (2 B/7 B) with our two-stage curriculum using LoRA. This adaptation and its strong CoIR results are our novel contributions.
> >
> > ---
> >
> > >“The paper talks about the sources of code data (Figure 2) but it does not detail the text data used in Stage 1.”
> >
> > As mentioned in lines 127–128, we use all training splits of BEIR text-to-text retrieval datasets (e.g., MS MARCO, TREC-COVID, FiQA, etc.). We will explicitly list these datasets in the revised manuscript for clarity.
> >
> > ---
> >
> > We appreciate your insights and hope this addresses your concerns. If there are specific references you’d like us to include, please let us know.
> >
> > Sincerely,
> > The CODEXEMBED Team

---

> > > ### Author Response · Authors · 2025-06-06
> > > **Seeking for Further Discussion**
> > >
> > > Dear Reviewer XpJL,
> > >
> > > We hope this message finds you well.
> > > We have posted detailed responses to each of the concerns you raised during the discussion phase. If anything remains unclear, please let us know—we would be glad to provide additional information or run further experiments before the discussion period ends.
> > >
> > > If our clarifications resolve your concerns, we would greatly appreciate your reconsideration of the current score.
> > >
> > > Thank you again for your thoughtful feedback and for your time during this busy review cycle.
> > >
> > > Best regards,
> > >
> > > The CodeXEmbed Team

---

### Decision · Program_Chairs · 2025-07-08

**Decision:**

Accept

**Comment:**

CodeXEmbed is a family of multilingual and multi-task code embedding models, ranging from 400 million to 7 billion parameters, designed to unify code and text retrieval within a single framework. Trained using a multi-stage LoRA-based pipeline across 12 programming languages and five retrieval tasks, CodeXEmbed achieves state-of-the-art performance in code retrieval, with its 7B model topping the CoIR benchmark and outperforming both open- and closed-source baselines. It also shows competitive results on the BEIR text retrieval benchmark (score: 60.06), while the smaller 2B and 400M variants surpass similarly sized models. Ablation studies demonstrate that using separate LoRA adapters with continuous optimization yields better performance. The paper further highlights that improvements in retrieval quality lead to better performance in downstream retrieval-augmented generation (RAG) tasks such as RepoEval and SWE-Bench-Lite, positioning CodeXEmbed as a strong foundation for code-centric RAG applications.

The work is interesting and valuable, particularly due to its support for both multilingual and multi-domain code retrieval. The two-stage training approach, using separate LoRA adapters, offers a slight but consistent improvement over the standard LoRA variant on the BEIR benchmark. The model's performance across BEIR text retrieval datasets appears to be state-of-the-art, and Table 1 presents a comprehensive evaluation across multiple settings. Overall, I find the contributions solid and the results convincing, and I would recommend the paper for acceptance.

Suggestions for Improvement:
- As suggested by a reviewer, please include the evaluation script along with a --seed flag in the released training code to enable reproducibility and support extensions for significance testing.
- As mentioned by the authors, in Section 3.4, more clearly highlight the gains from the two-stage separate-LoRA approach to better emphasize its impact. To further support reproducibility, consider releasing the model checkpoints and scripts used to produce the BEIR and CoIR results.